# PET/MRI and Novel Targets for Breast Cancer

**DOI:** 10.3390/biomedicines12010172

**Published:** 2024-01-12

**Authors:** Hyun Woo Chung, Kyoung Sik Park, Ilhan Lim, Woo Chul Noh, Young Bum Yoo, Sang Eun Nam, Young So, Eun Jeong Lee

**Affiliations:** 1Department of Nuclear Medicine, Konkuk University Medical Center, Konkuk University School of Medicine, 120-1 Neungdong-ro, Gwangjin-gu, Seoul 05030, Republic of Korea; hwchung@kuh.ac.kr (H.W.C.); youngso@kuh.ac.kr (Y.S.); 2Department of Surgery, Konkuk University Medical Center, Konkuk University School of Medicine, 120-1 Neungdong-ro, Gwangjin-gu, Seoul 05030, Republic of Korea; 20210415@kuh.ac.kr (W.C.N.); 0117652771@kuh.ac.kr (Y.B.Y.); 20090055@kuh.ac.kr (S.E.N.); 3Research Institute of Medical Science, Konkuk University School of Medicine, 120-1 Neungdong-ro, Gwangjin-gu, Seoul 05030, Republic of Korea; 4Department of Nuclear Medicine, Korea Cancer Center Hospital, Korea Institute of Radiological and Medical Sciences (KIRAMS), 75 Nowon-ro, Nowon-gu, Seoul 07812, Republic of Korea; erosion@hanmail.net; 5Department of Nuclear Medicine, Seoul Medical Center, 156 Sinnae-ro, Jungnang-gu, Seoul 02053, Republic of Korea; cateunjeong@naver.com

**Keywords:** breast cancer, PET/MRI, estrogen receptor, HER2, FAPI, hypoxia

## Abstract

Breast cancer, with its global prevalence and impact on women’s health, necessitates effective early detection and accurate staging for optimal patient outcomes. Traditional imaging modalities such as mammography, ultrasound, and dynamic contrast-enhanced magnetic resonance imaging (MRI) play crucial roles in local-regional assessment, while bone scintigraphy and ^18^F-fluorodeoxyglucose positron emission tomography/computed tomography (^18^F-FDG PET/CT) aid in evaluating distant metastasis. Despite the proven utility of ^18^F-FDG PET/CT in various cancers, its limitations in breast cancer, such as high false-negative rates for small and low-grade tumors, have driven exploration into novel targets for PET radiotracers, including estrogen receptor, human epidermal growth factor receptor-2, fibroblast activation protein, and hypoxia. The advent of PET/MRI, which combines metabolic PET information with high anatomical detail from MRI, has emerged as a promising tool for breast cancer diagnosis, staging, treatment response assessment, and restaging. Technical advancements including the integration of PET and MRI, considerations in patient preparation, and optimized imaging protocols contribute to the success of dedicated breast and whole-body PET/MRI. This comprehensive review offers the current technical aspects and clinical applications of PET/MRI for breast cancer. Additionally, novel targets in breast cancer for PET radiotracers beyond glucose metabolism are explored.

## 1. Introduction

Breast cancer is the cancer with the highest incidence and mortality among women worldwide [1]. Early detection and precise staging are crucial for effective treatment and improved patient outcomes [2]. Mammography, ultrasound (US), and dynamic contrast-enhanced (DCE) magnetic resonance imaging (MRI) are well-established local-regional imaging methods for breast cancer. Bone scintigraphy and ^18^F-fluorodeoxyglucose positron emission tomography/computed tomography (^18^F-FDG PET/CT) are also available to evaluate distant metastasis [3].

^18^F-FDG PET/CT has been widely validated for various cancers, playing an important role in diagnosis, initial staging, therapy response evaluation, and restaging [4]. However, its application for breast cancer in terms of detection, differential diagnosis of benign from malignant lesions, and local tumor staging is not recommended. This is due to its high false-negative rate for small (<1 cm) and low-grade breast cancer, a high false-positive rate for local benign breast disease, and low sensitivity for the detection of axillary lymph node metastasis [5]. Consequently, researchers have explored various novel PET radiotracers targeting molecular factors. Examples include ^18^F-fluoroestradiol (FES) for estrogen receptor (ER), ^89^Zr-Df-trastuzumab for human epidermal growth factor receptor-2 (HER2), ^68^Ga-fibroblast-activation-protein-inhibitor (FAPI) for fibroblast activation protein (FAP), and ^18^F-fluoromisonidazole (FMISO) for hypoxia. In May 2022, the United States Food and Drug Administration approved ^18^F-FES for patients with recurrent and metastatic breast cancer, serving as an adjunct to biopsy for detecting ER-positive lesions [3].

PET/MRI, which combines the metabolic information of PET with the high anatomical details of MRI, has been suggested as a promising synergistic imaging modality for cancer. Since MRI is highly sensitive for breast cancer and does not expose the patient to radiation, PET/MRI is under active investigation across the spectrum of diagnosis, staging, treatment response assessment, and restaging of breast cancer [6].

This review article offers comprehensive insights into the current technical aspects and clinical applications of PET/MRI for breast cancer. Additionally, novel targets in breast cancer for PET radiotracers beyond glucose metabolism are explored, describing their mechanisms of action and potential impact on the management of patients with breast cancer.

## 2. Technical Aspects of PET/MRI

### 2.1. Integration of PET and MRI

Since photomultiplier tubes (PMTs) used to detect photons from their scintillation crystal are highly susceptible to magnetic fields, the integration of standard PET and MRI together was technically challenging. As a result, the initial combined PET and MRI system was installed as separate PET/CT and MRI systems with co-registration and fusion of images [7]. With the advancement of light detectors operable in a magnetic field, such as avalanche photodiodes (APDs) and silicon photomultipliers (SiPMs), simultaneous acquisition of PET and MRI images has been achieved.

The first fully integrated PET/MRI system was designed with an APD-based PET system [8]. However, the limited timing resolution of the APDs did not allow the time of flight (TOF) measurement, a capability available in PMT-based PET detector systems in modern PET/CT [9]. Subsequently, a TOF-capable fully integrated PET/MRI system was developed using SiPMs as light detectors [10]. SiPM-based detectors, also utilized in state-of-the-art digital PET/CT systems, provide each scintillation crystal coupled to a single SiPM (1:1 coupling) and convert scintillation light directly to a digital signal (output 1 for detected photons, output 0 for no photon detected). This reduces the need for amplification to produce a summed analog signal and decreases noise [11]. Furthermore, enhanced TOF measurement and reduced dead time also improve spatial and timing resolutions [12]. 

### 2.2. Patient Preparation

Basically, instructions to patients for PET imaging in PET/MRI are similar to the conventional PET/CT recommendations, depending on the radiotracer used. However, due to the high sensitivity and efficiency of SiPM-based PET detectors, PET/MRI can achieve substantially lower radiation exposure. This reduction is not only due to omitting CT but also involves reducing the injected dose of the radiotracer, especially for ^18^F-FDG [13].

All safety regulations and guidelines related to conventional MRI safety equally apply to PET/MRI [14]. This includes screening for potential MRI safety contraindications, such as claustrophobia, passive and active implants, metallic inclusions, and pregnancy. Of note, all current PET/MRI systems operate at a 3-Tesla static magnetic field strength. This can be particularly practical for ensuring MRI safety when planning to scan patients with MRI conditional cardiac pacemakers or other active implants.

### 2.3. Image Acquisition

Patients are first injected with the selected radiotracer. After the tracer has distributed within the body, PET and multiparametric MRI are performed simultaneously in the prone position, with both breasts positioned in the dedicated breast coil. PET/MRI scanners have utilized 4-, 8-, and 16-channel radiofrequency coils for breast imaging. Since the administration of MRI contrast agents is not influenced by simultaneous PET acquisition, the MRI imaging protocol follows the standard breast MRI protocol of the institution. This includes conventional sequences such as T2-weighted imaging with or without fat suppression, diffusion-weighted imaging (DWI), and T1-weighted DCE imaging [15]. However, MR-based attenuation correction (AC) for PET imaging, mostly performed using the standard Dixon-based AC method, must be completed before initiating MRI contrast injection to prevent faulty tissue segmentation due to signal intensity changes in the MRI images [16].

DWI is performed either before or after contrast agent administration, as its timing does not affect apparent diffusion coefficient (ADC) calculations. However, post-contrast ADC parameters may be slightly lower than pre-contrast ADC parameters due to susceptibility artifacts, making pre-contrast acquisition preferable [17]. DCE-MRI is acquired after the intravenous injection of a contrast agent, following either routine or fast/ultrafast protocols. The total acquisition time of dedicated breast ^18^F-FDG PET/MRI is recommended to be as minimal as possible [18].

Whole-body PET/MRI is usually performed after breast PET/MRI, using the previously injected radiotracer and an MRI contrast agent. The patient is positioned supine with surface radiofrequency coils, and PET data and MRI sequences are acquired concurrently at each bed position. The MRI protocol includes T2-weighted images, DWI, and T1-weighted post-contrast sequences. Recently, fast breast and whole-body MRI protocols have been proposed for patient comfort [19,20]. A summary of breast and whole-body PET/MRI protocols is illustrated in Figure 1.

## 3. ^18^F-FDG PET/MRI for Breast Cancer

### 3.1. Diagnosis

Breast MRI is the most sensitive modality for detecting breast cancer [21]. However, the moderately specific nature of breast MRI results in false-positive findings, necessitating additional imaging and biopsy [22]. Thus, reducing false-positive findings from breast MRI has significant potential impact by avoiding additional biopsies, reducing cost, decreasing patient anxiety, and minimizing time to surgery. 

^18^F-FDG PET has a limited sensitivity in small breast lesion with both false-negative and false-positive findings as a benign lesion can have an increased uptake. However, the addition of ^18^F-FDG PET with multiparametric MRI has been suggested to increase the specificity, especially when the size of breast lesions is more than 10 mm [23]. Furthermore, the addition of dedicated prone breast PET/MRI to supine whole-body imaging is reported to be more sensitive than whole-body-only supine imaging. In a study involving 38 prospectively enrolled patients with 56 breast cancer lesions, dedicated prone breast ^18^F-FDG PET/MRI, combined with supine whole-body imaging, correctly identified breast cancers in 97% of cases (37/38). In contrast, supine whole-body-only imaging missed five patients (87%, 33/38) [24]. The unidentified patient in dedicated prone breast ^18^F-FDG PET/MRI had a pT1a tumor measuring 5 mm without radiotracer uptake and considered a benign lesion. The missed five patients in supine whole-body-only ^18^F-FDG PET/MRI had small lesions ranging from 5 to 13 mm. A meta-analysis by Ruan et al. with ten lesion-based studies and three patient-based studies showed good performance of ^18^F-FDG PET/MRI in diagnosing breast cancer [25]. For a lesion-based dataset, the pooled sensitivity, specificity, and area under the curve (AUC) were 95%, 91%, and 0.96, respectively, and for a patient-based dataset, the pooled sensitivity, specificity, and AUC were 97%, 97%, and 1.00, respectively.

With recent advances in artificial intelligence (AI) and its applications in medical imaging, Romeo et al. demonstrated that the AI-based radiomics model, with features extracted from simultaneous multiparametric ^18^F-FDG PET/MRI, achieved high accuracy in discriminating between benign and malignant breast lesions, with an AUC of 0.983 [26]. Its sensitivity was not statistically different from the clinical interpretation by experts (100% for AI-based radiomics model vs. 95.3% for clinical interpretation), but specificity was higher for the AI-based radiomics model (94.3% vs. 73.7%, respectively), indicating the potential to decrease false-positive findings in benign breast lesions.

While ^18^F-FDG PET/MRI is not currently recommended for breast cancer diagnosis, its utilization could improve the diagnostic accuracy of MRI and potentially enable a less invasive, comprehensive diagnostic strategy.

### 3.2. Initial Staging

The American Joint Committee on Cancer (AJCC) anatomic TNM staging system for breast cancer includes the extent of the tumor (T), the spread to regional lymph nodes (N), and distant metastasis (M). It emphasizes that comprehensive initial anatomic staging using mammography, US, and MRI is crucial for guiding patient treatment decisions [27]. 

MRI has demonstrated greater accuracy than conventional imaging methods in assessing the extent of breast tumors [28]. Consequently, ^18^F-FDG PET/MRI is proposed to be superior for T-staging compared to conventional mammography, ultrasound (US), and ^18^F-FDG PET/CT, and is, at the very least, equivalent to breast MRI alone. In a study by Grueneisen et al., involving 49 patients with 83 biopsy-proven invasive breast cancers, mostly at more than T1c stage, no significant difference was observed in correct T-staging between ^18^F-FDG PET/MRI and MRI. However, both modalities were significantly more accurate than ^18^F-FDG PET/CT (PET/MRI and MRI, 82%; PET/CT, 68%) [29]. In another study by Goorts et al., conducted on 40 patients with breast cancer before neoadjuvant chemotherapy, multifocal tumors were identified in four patients after MRI. However, ^18^F-FDG PET/MRI did not reveal additional multifocal tumors, indicating that the clinical T stage, based on the primary tumor size and the number of lesions, did not differ between ^18^F-FDG PET/MRI and MRI alone [30].

In N staging, early studies initially reported the performance of ^18^F-FDG PET/MRI as either equivalent to or inferior to MRI alone [29,31]. However, Morawitz et al. demonstrated the diagnostic superiority of ^18^F-FDG PET/MRI over MRI and CT in determining the regional lymph node status. Their prospective double-center study involved 182 patients with newly diagnosed, treatment-naïve breast cancer [32]. ^18^F-FDG PET/MRI detected significantly more nodal-positive patients than MRI and CT. Moreover, across all lymph node stations (axillary, supraclavicular, and internal mammary stations), ^18^F-FDG PET/MRI identified significantly more lymph node metastases compared to MRI and CT. Consequently, ^18^F-FDG PET/MRI resulted in nodal upstaging in 30 patients compared to MRI and in 41 patients compared to CT. No downstaging occurred in ^18^F-FDG PET/MRI compared to MRI or CT. MRI upstaged nodes in 15 patients and downstaged nodes in 6 patients compared to CT. Additionally, simple imaging features from ^18^F-FDG PET/MRI and MRI can be utilized for N staging in patients with newly diagnosed breast cancer through machine-learning–based prediction models, exhibiting high accuracy [33]. The diagnostic accuracy for MRI features was 87.5% for both the machine-learning algorithm and radiologists. For ^18^F-FDG PET/MRI, the diagnostic accuracy was 91.2% and 89.3% for the machine-learning algorithm and radiologists, respectively, with no significant difference. An example of ^18^F-FDG PET/MRI for initial N staging in a patient with breast cancer is shown in Figure 2.

Currently, ongoing clinical trials are evaluating the axillary staging performance of ^18^F-FDG PET/MRI compared to sentinel node biopsy (SNB) in both advanced and early breast cancers [34]. These trials consist of two prospective comparative single-center studies conducted in different settings. In the first trial (ClinicalTrials.gov Identifier: NCT04826211), the staging performance of ^18^F-FDG PET/MRI is compared with SNB in patients with breast cancer undergoing primary systemic therapy (PST) for positive axillary lymph nodes at diagnosis. Initial staging and post-PST ^18^F-FDG PET/MRI results, which are used to plan surgery, will be compared with the results of SNB. In the second trial (ClinicalTrials.gov Identifier: NCT04829643), the axillary staging of ^18^F-FDG PET/MRI is compared with SNB in patients with early breast cancer undergoing surgery. The results from these trials have the potential to offer patients a less invasive and de-escalated axillary surgery with improved outcomes.

A recent study compared conventional imaging (contrast-enhanced CT, axillary US, and bone scintigraphy), MRI, and ^18^F-FDG PET/MRI to evaluate N and M staging in 208 patients with newly diagnosed breast cancer [35]. ^18^F-FDG PET/MRI detected more significant regional lymph node metastases and provided more accurate estimates of the clinical N stage than conventional imaging and MRI. Regarding the M staging, there was a trend toward higher sensitivity for ^18^F-FDG PET/MRI and MRI compared to conventional imaging in detecting distant metastasis. However, no significant differences were found between the various imaging modalities.

In the comparison of ^18^F-FDG PET/MRI and ^18^F-FDG PET/CT for M staging of breast cancer, higher sensitivity and lower specificity of ^18^F-FDG PET/MRI were generally found, particularly for osseous and/or hepatic metastases [36]. Recent meta-analysis, including 16 articles involving 1261 patients, indicated that ^18^F-FDG PET/MRI showed superior sensitivity and similar specificity to ^18^F-FDG PET/CT in detecting bone metastases in patients with breast cancer [37]. Another meta-analysis with a subgroup using three studies (182 patients) reported that ^18^F-FDG PET/MRI had higher sensitivity and specificity for detecting distant metastasis of breast cancer than ^18^F-FDG PET/CT [38]. The pooled sensitivity, specificity, and AUC for ^18^F-FDG PET/MRI were 95%, 96%, and 0.98, respectively. For ^18^F-FDG PET/CT, the pooled sensitivity, specificity, and AUC were 87%, 94%, and 0.94, respectively. An example of ^18^F-FDG PET/MRI for initial M staging in a patient with breast cancer is shown in Figure 3.

### 3.3. Therapy Response Assessment

While the assessment of tumor response after therapy still relies on changes in size, ^18^F-FDG PET/MRI shows greater potential than conventional anatomical imaging. ^18^F-FDG PET/MRI can provide functional data such as metabolism (PET), cell proliferation (DWI), and neoangiogenesis (DCE-MRI), along with anatomical details, in the assessment and early prediction of systemic therapy response. 

An early study with nine patients by Jena et al. reported that PET metabolic parameters, such as maximum standardized uptake value (SUVmax), and DCE-MRI pharmacokinetic parameters, such as K^trans^ (volume transfer constant between blood plasma and interstitial space), obtained from simultaneous ^18^F-FDG PET/MRI were reduced after chemotherapy in patients who have responded to therapy [39]. Wang et al. suggested that changes in combined PET and MRI parameters, such as SUVmax, total lesion glycolysis (TLG), and minimum ADC (ADCmin), in sequential ^18^F-FDG PET/MRIs obtained before and after the first or second cycle of neoadjuvant chemotherapy (NCT), predict treatment response more accurately than individual PET/MRI parameters [40]. Cho et al. demonstrated that TLG obtained from PET and signal enhancement ratio (SER) from MRI in ^18^F-FDG PET/MRI predicted non-pathological complete response (pCR) after the first cycle of NCT in patients with breast cancer [41]. Sekine et al. evaluated the performance of ^18^F-FDG PET/MRI, mammography, and US in 74 patients with breast cancer in prediction of pCR after NCT [42]. The prediction of pCR by ^18^F-FDG PET/MRI depended on the absence of detectable enhancement on MRI and/or lack of meaningful uptake on PET. The overall sensitivity of ^18^F-FDG PET/MRI, mammography, and US were 72%, 71%, and 17%, respectively. The overall specificity of ^18^F-FDG PET/MRI, mammography, and US were 79%, 80%, and 91%, respectively. In another study, de Mooij et al. conducted a prospective study involving 41 patients with 42 primary invasive breast cancers who underwent NCT before surgery [43]. Qualitative evaluation using ^18^F-FDG PET/MRI after NCT predicted the therapy response in the primary tumor but not the response in axillary lymph node metastasis. When compared to qualitative evaluation after NCT, combining quantitative variables such as decreased SUVmax and SER in the primary tumor and axillary lymph node metastasis from sequential ^18^F-FDG PET/MRI during NCT improved the evaluation and prediction of response to NCT. Moreover, deep learning techniques and radiomics-based analysis may be helpful in predicting the response to NCT using ^18^F-FDG PET/MRI [44,45].

### 3.4. Restaging

Early detection and characterization of local-regional recurrence and distant metastasis are essential for optimal treatment and prognosis in patients with previously treated breast cancer. Curative surgery or radiation therapy may be available for local-regional recurrence, whereas palliative systemic therapy is necessary for distant metastases [46]. 

^18^F-FDG PET/CT has been recommended as a helpful imaging modality in situations where standard staging studies are equivocal or suspicious [5]. In a prospective study by Sawiki et al., 21 patients with 134 suspected breast cancer recurrent lesions underwent whole-body PET/CT with iodinated contrast. Subsequently, they underwent PET/MRI with a gadolinium-based contrast agent. PET/CT and PET/MRI were performed in a single injection of ^18^F-FDG [47]. For patient-based analysis, ^18^F-FDG PET/MRI, ^18^F-FDG PET/CT, and MRI_PET/MRI_ correctly identified all 17 patients with cancer recurrence. CT_PET/CT_ identified 15 of the 17 patients correctly (88.2%). In lesion-based analysis, ^18^F-FDG PET/MRI, ^18^F-FDG PET/CT, MRI_PET/MRI_, and CT_PET/CT_ correctly identified 98.5% (132/134), 94.8% (127/134), 88.1% (118/134), and 57.5% (77/134) of all lesions, respectively. Notably, bone, lymph node (<10 mm), and liver metastases were missed by images, particularly by CT_PET/CT_. Interobserver agreement was substantial for both ^18^F-FDG PET/MRI and ^18^F-FDG PET/CT, moderate for MRI_PET/MRI_, and fair for CT_PET/CT_. A meta-analysis, including eight studies, found that ^18^F-FDG PET/MRI performed well for staging/restaging in primary, metastatic, and recurrent breast cancer [48]. Patient-based overall pooled sensitivity, specificity, and AUC were 98%, 87%, and 0.99, respectively. Lesion-based overall pooled sensitivity, specificity, and AUC were 91%, 95%, and 0.99, respectively. The overall diagnostic accuracies were 96% (patient-based analysis) and 95% (lesion-based analysis). An example of ^18^F-FDG PET/MRI for restaging in a patient with previously treated breast cancer is shown in Figure 4.

## 4. Novel Targets beyond Glucose Metabolism

### 4.1. Estrogen Receptor (ER)

The ER, expressed in approximately 70% of breast cancers at presentation, is a significant steroid hormone receptor for female physiology and carcinogenesis. It is also crucial as an effective therapeutic target. Pathologic assessment of ER status by tissue assay, commonly using immunohistochemistry methods, is essential to guide the ER-targeted systemic therapy of breast cancer [50].

According to the practice guideline for ER imaging of patients with breast cancer using ^18^F-FES PET, approved in June 2023 by Society of Nuclear Medicine and Molecular Imaging (SNMMI)/European Association of Nuclear Medicine (EANM), common clinical indications include: (1) assessing lesions that are difficult to biopsy or yield non-diagnostic results after biopsy, (2) guiding therapy after the progression of metastases, (3) guiding therapy at the initial presentation of metastases, and (4) detecting cancer with ER expression when other imaging tests are equivocal or suspicious for cancer [51,52]. Other emerging indications under investigation are as follows: (1) detecting ER-expressing lesions in patients with suspected or known recurrent or metastatic breast cancer, (2) assessing ER status, instead of biopsy, in lesions that are easily accessible for biopsy, (3) staging invasive lobular breast cancer and low-grade ER-expressing invasive ductal cancer, (4) routinely staging extra-axillary nodes and distant metastases to evaluate ER-expression [53].

Compared with ^18^F-FDG, ^18^F-FES does not accumulate at sites of inflammation or degeneration [54]. This property allows differentiation between ER-positive cancer/metastasis and other findings, such as inflammatory lymph nodes, degenerative osseous lesions, or reactive bone marrow. If ^18^F-FES PET yields a positive result for an indeterminate finding on a prior ^18^F-FDG PET, it can be assessed as ER-positive breast cancer/metastasis, supporting treatment decisions. In cases where ^18^F-FES PET shows a negative result for an indeterminate ^18^F-FDG PET finding, various differential diagnoses, such as benign etiologies, ER-negative breast cancer/metastasis, and other malignant etiologies, should be considered [55].

For patient preparation, unlike ^18^F-FDG, ^18^F-FES is not significantly affected by dietary state or exercise for uptake. Thus, fasting and avoidance of strenuous exercise before imaging are not required [53]. A clinical history of any previous treatment that blocks the ER and reduces the uptake of ^18^F-FES, such as tamoxifen and fulvestrant, should be provided before the ^18^F-FES PET. The withdrawal of ER antagonists is recommended for at least six weeks. However, since ER-targeting drugs have various half-lives and washout times, more data may be necessary to estimate the appropriate withdrawal period [51].

### 4.2. Human Epidermal Growth Factor Receptor-2 (HER2)

HER2 is a membrane protein encoded by the erythroblastic oncogene B (*ERBB2*). *ERBB2* amplification is found in 10–15% of invasive breast carcinomas and classified as HER2-positive by immunohistochemistry. HER2 is considered as HER2-zero when no detectable levels are present and HER2-low when levels are low. HER2-positive breast cancers are indicated to anti-HER2 therapies [56]. The ability to noninvasively detect HER2 overexpression through imaging is expected to have a significant clinical impact. This is especially crucial in patients with multifocal/multicentric tumors, where assessment using core biopsy can be challenging. Ulaner et al. demonstrated that HER2-targeted PET can detect HER2-positive metastases in patients with HER2-negative primary breast cancer, meaning that HER2 PET can identify additional candidates for HER2-targeted therapy [57]. Additionally, HER2 PET can be useful to evaluate the status of HER2 expression in non-responders when HER2 status cannot be determined with the standard workup [58]. An example of ^64^Cu-DOTA-trastuzumab PET image in a patient with breast cancer is shown in Figure 5 [59].

Until now, anti-HER2 therapies were not indicated for HER2-low or HER2-zero breast cancers. However, the recently conducted DESTINY-Breast04 trial, Trastuzumab Deruxtecan (T-DXd) in Previously Treated HER2-Low Advanced Breast Cancer, led by Modi et al., reported that T-DXd resulted in significantly longer progression-free and overall survival than the physician’s choice of chemotherapy in patients with HER2-low metastatic breast cancer [60]. As the current standard immunohistochemistry assays were not precisely standardized to distinguish between HER2-low and HER2-zero patients, HER2 PET could serve as a promising alternative modality [61]. It enables the quantitative and noninvasive monitoring of whole-body HER2 expression for novel HER2-targeted treatments. Therefore, while patients with HER2-zero metastatic breast cancer are not eligible for T-DXd, HER2 PET could visualize some HER2-low lesions in these patients, making them possibly eligible for T-DXd. Additionally, HER2 PET could be a marker for predicting the efficacy of T-DXd among patients who are currently eligible for this treatment by identifying those who do not benefit from T-DXd.

### 4.3. Fibroblast Activation Protein (FAP)

The tumor microenvironment (TME) with a strong desmoplastic reaction has been recognized as a contributor to tumor progression and resistance to therapy [62]. Cancer-associated fibroblasts (CAFs) play a significant role in the structural organization of the TME. They are heterogeneous and highly abundant in desmoplastic tumors and are also observed in various solid tumors. Fibroblast activation protein (FAP), a type-II transmembrane serine protease not normally expressed in healthy tissue but in pathological conditions such as fibrosis, arthritis, and cancer, has been investigated as a marker for targeting CAFs [63]. 

Since fibroblast activation protein inhibitor (FAPI) has emerged as a therapeutic target in various malignancies including breast cancer, FAPI-based radiopharmaceuticals labeled for imaging and therapy have been evaluated [64]. In 2018, FAP-targeting PET radiopharmaceuticals such as ^68^Ga-FAPI, which allow labeling with not only ^68^Ga but also therapeutic radioisotopes such as ^177^Lu, were introduced [65]. Compared with ^18^F-FDG PET, FAPI PET is not associated with glucose metabolism. This lack of association results in reduced background activity in the brain, liver, oro- and naso-pharyngeal mucosa, or gastrointestinal tract, leading to an improved tumor-to-background ratio (TBR) and image contrast [66].

^68^Ga-FAPI-04 and ^68^Ga-FAPI-46 PET imaging demonstrated favorable results for the diagnosis, staging, and response assessment to NCT in patients with invasive breast cancer [67,68,69]. Backhaus et al. reported that ^68^Ga-FAPI-46 PET/MRI showed strong uptake in all 18 untreated primary invasive breast cancers, even for subcentimeter lesions. For axillary lymph node staging, strong uptake was found in all 13 patients with lymph node metastases. PET/MRI also detected extra-axillary LN involvement in seven patients and affected therapy decisions in three patients [68]. In a later report by Backhaus et al., ^68^Ga-FAPI-46 PET/MRI could classify breast response to NCT correctly in all 13 patients using readers’ visual assessment or TBR ratio. Evaluation of MRI alone resulted in at least two false-positives. For lymph nodes, PET/MRI had at least two false-negatives, whereas MRI alone resulted in two false-negatives and one false-positive [69]. Anticipating the clinical advantages for easier availability with ^18^F compared to ^68^Ga, ^18^F-A1F-FAPI-74 was also investigated [70]. 

For therapeutic purposes, Lindner et al. reported that two patients with metastasized breast cancer with high ^68^Ga-FAPI-04 uptake in metastases had a reduction in pain symptoms after therapy with a considerably low dose of ^90^Y-FAPI-04 [71]. In particular, FAPI-46 demonstrated an increasing TBR over time, making it a promising radiotracer for future theranostic application [72]. Recently, Yadav et al. suggested that ^177^Lu-DOTAGA-FAPI dimer treatment is well-tolerated and effective for patients with end-stage metastatic breast cancer [73]. The results showed 26.3% complete response, 15.7% partial response, 42% minimal response, 11% stable disease, and 5% no response, all without adverse events of grade 3 or 4. 

### 4.4. Hypoxia

Tumor hypoxia, resulting from abnormal vascularity that limits the oxygen demand for tumor growth, drives tumors to develop adaptive responses to low-oxygen conditions. This leads to altered gene expression and cell metabolism, contributing to their survival in an unfavorable environment [74]. Thus, tumor hypoxia causes malignant reprogramming for metabolic adaptation, invasion, metastasis, and angiogenesis, which is related to increased metastasis and invasion, poor response to radiotherapy and chemotherapy, and poor prognosis [75]. Since the introduction of ^18^F-fluoromisonidazole (^18^F-FMISO) in 1986, most PET radiotracers for imaging tumor hypoxia belong to the 2-nitroimidazole compound family [76]. The next-generation of 2-nitroimidazoles with lower lipophilicity was developed to provide higher image contrast by faster clearance from non-target tissues compared to ^18^F-FMISO [77].

Carmona-Bozo et al. reported the association between hypoxia and vascular function in patients with treatment-naïve breast cancer using simultaneous ^18^F-FMISO PET/MRI [78]. Tumor hypoxia measured by ^18^F-FMISO PET and markers of perfusion and vascular function from DCE-MRI showed an inverse relationship, which supports the hypothesis of perfusion-driven hypoxia in breast cancer. They suggested that the combined hypoxia-perfusion status of tumors might be considered to determine the treatment method and predict efficacy in breast cancer. This emphasizes the importance of simultaneous multimodality imaging.

## 5. Discussion

Compared with traditional imaging methods and PET/CT, PET/MRI may offer advantages for anatomical definition, particularly in certain parts of the body, including the brain, neck, breast, liver, pelvis, and bone. Diagnostic confidence can be increased with the complementary information provided by PET/MRI. Furthermore, dedicated prone breast PET/MRI combined with supine whole-body imaging offers more valuable information for evaluating breast cancer than supine whole-body-only imaging.

In a prospective cohort study of 56 patients with newly diagnosed and therapy-naïve breast cancer from two university hospitals, Kirchiner et al. suggested that staging by ^18^F-FDG PET/MRI resulted in a difference in treatment compared to the traditional staging algorithm (mammography, breast US, chest plain radiography, bone scintigraphy, US of the liver and axillary fossa, and follow-up with CT and/or MRI) in 8/56 patients (14%) [79]. For example, while the primary tumor was regarded as T2 by mammography/US, PET/MRI showed infiltration of the skin, which was confirmed by histopathology later, and the primary tumor was rated as T4. Therefore, a mastectomy and radiotherapy of the chest wall were recommended instead of breast-conserving therapy without radiation. Regarding the therapy recommendation of regional lymph nodes, differences between the traditional staging algorithm and the ^18^F-FDG PET/MRI were present in 6/56 patients (11%). In these patients, axillary dissection was recommended for lymph nodes in 5 cases (83%) instead of SNB based on the ^18^F-FDG PET/MRI that were not visible in US. As a result, four cases were histopathologically confirmed as malignant. However, in one other case, ^18^F-FDG PET/MRI had a false-negative result for axillary lymph node metastasis compared with traditional staging algorithm. A trend toward higher staging by ^18^F-FDG PET/MRI was found without statistical significance.

Since PET/MRI is used less commonly in current clinical practice than PET/CT, most studies reviewed in this article regarding novel PET radiotracers for breast cancer beyond ^18^F-FDG were conducted with PET/CT rather than PET/MRI. Additional studies are needed to provide evidence of the benefits of using PET/MRI, instead of PET/CT, for the novel PET radiotracers in the management of breast cancer.

## 6. Conclusions

PET/MRI is a multimodal imaging technique that integrates metabolic and anatomical information. Recent technical advancements have enabled the simultaneous acquisition of PET and MRI, enhancing diagnostic accuracy. Consequently, ^18^F-FDG PET/MRI has demonstrated potential in the diagnosis, initial staging, therapy response assessment, and restaging of patients with breast cancer. The ongoing exploration of novel targets for PET radiotracers beyond glucose metabolism shows promise for more targeted approaches to breast cancer diagnosis and treatment. Further investigations and a better understanding are necessary to define the role and optimal patient population for PET/MRI and novel radiotracers in breast cancer.

## Figures and Tables

**Figure 1 biomedicines-12-00172-f001:**
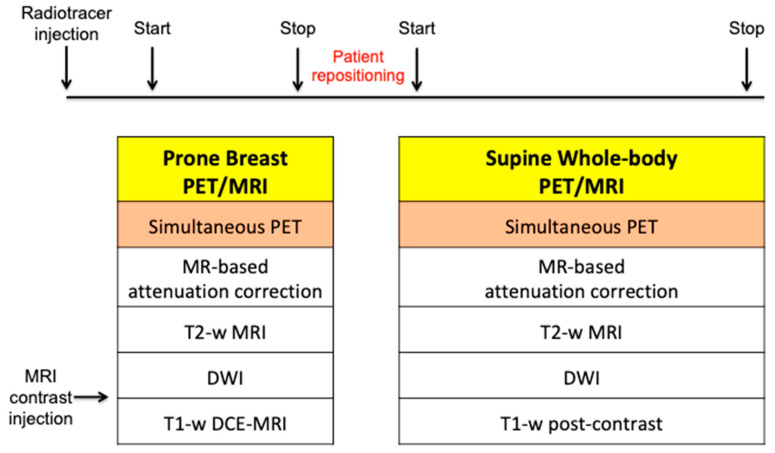
Breast and whole-body simultaneous PET/MRI acquisition protocol.

**Figure 2 biomedicines-12-00172-f002:**
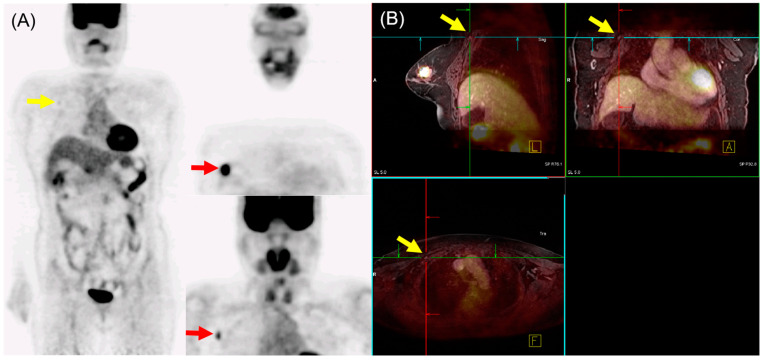
^18^F-FDG PET/MRI for initial N staging in a patient with breast cancer. (**A**) PET images revealed hypermetabolic right breast cancer with right axillary lymph node metastasis (red arrow). Additionally, inconclusive mild hypermetabolic uptake was found in the right interpectoral area (yellow arrow). (**B**) PET/MRI identified right interpectoral lymph node metastasis with gadolinium-contrast enhancement on T1-w MRI (yellow arrow).

**Figure 3 biomedicines-12-00172-f003:**
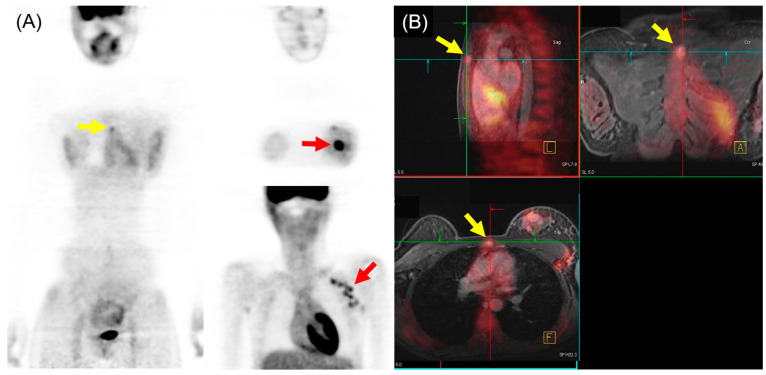
^18^F-FDG PET/MRI for initial M staging in a patient with breast cancer. (**A**) PET images revealed hypermetabolic left breast cancer with left axillary lymph node metastases (red arrow). Additionally, inconclusive mild hypermetabolic uptake was observed in the sternum area (yellow arrow). (**B**) PET/MRI identified sternum metastasis with gadolinium-contrast enhancement on T1-w MRI (yellow arrow).

**Figure 4 biomedicines-12-00172-f004:**
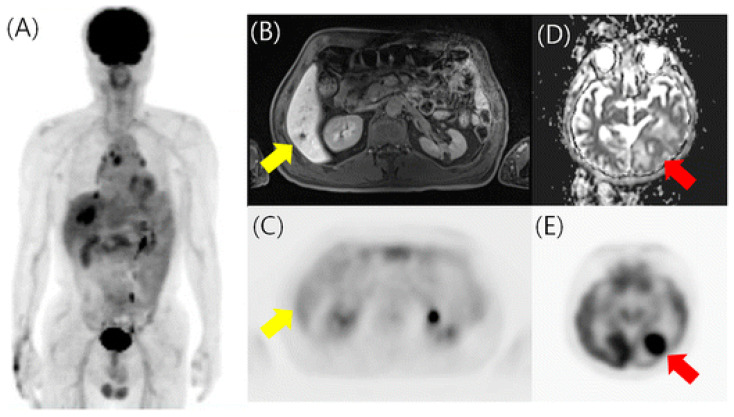
^18^F-FDG PET/MRI in a patient with previously treated breast cancer. (**A**) PET maximum intensity projection image revealed hepatic metastases. (**B**,**C**) Post-contrast MRI found an additional hepatic metastasis without hypermetabolism on PET (yellow arrow). (**D**,**E**) MRI ADC and PET images detected additional asymptomatic brain metastasis (red arrow). Reprinted through the Copyright Clearance Center’s RightsLink^®^ service from Kwon et al. [49].

**Figure 5 biomedicines-12-00172-f005:**
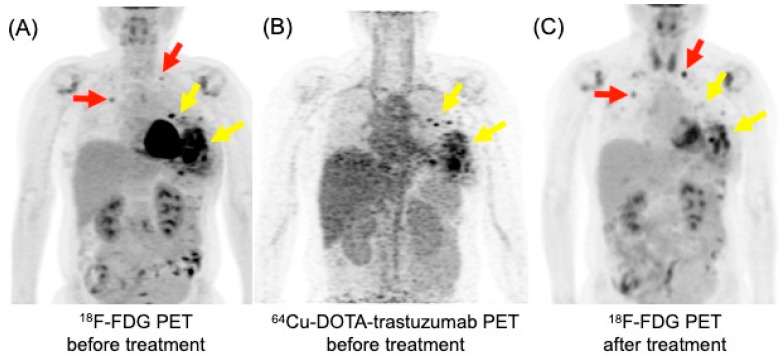
^18^F-FDG and ^64^Cu-DOTA-trastuzumab PET maximum intensity projection images in a patient with recurrent left breast cancer with pulmonary metastases. (**A**) ^18^F-FDG PET detected hypermetabolic recurrent left breast cancer (yellow arrow) and pulmonary metastases (red arrow). (**B**) ^64^Cu-DOTA-trastuzumab PET demonstrated increased uptake only in the recurrent left breast lesions (yellow arrow), not in the pulmonary metastases. (**C**) Following treatment with trastuzumab emtansine, the metabolic activity of the recurrent left breast lesions decreased (yellow arrow), while the pulmonary metastases worsened on ^18^F-FDG PET (red arrow).

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
