# Peer review of "PET/MRI and Novel Targets for Breast Cancer"

_biomedicines, 2024, doi:10.3390/biomedicines12010172_

Round 1

Reviewer 1 Report

Comments and Suggestions for Authors

The topic of this article has considerable clinical significance and highlights the role of hybrid imaging with PET/MRI in the diagnosis and treatment of patients with breast cancer, the most common type of cancer worldwide.

As the utility of PET/MRI in many clinical settings is still questionable, this article provides a comprehensive and detailed account of the utility of this hybrid scanner in breast cancer patients. 

However, I have a few minor comments, mainly concerning grammar: I have a few minor comments, mainly concerning grammar: 

- Page 2 - " Patient Preparation": please use " On note" instead of "By the way". 

- Page 5 - " Therapy Response Assessment", 2nd paragraph: Please replace "in responders" with "in patients who have responded to therapy". 

Author Response

The topic of this article has considerable clinical significance and highlights the role of hybrid imaging with PET/MRI in the diagnosis and treatment of patients with breast cancer, the most common type of cancer worldwide.

As the utility of PET/MRI in many clinical settings is still questionable, this article provides a comprehensive and detailed account of the utility of this hybrid scanner in breast cancer patients.

However, I have a few minor comments, mainly concerning grammar:

  1. Page 2 - " Patient Preparation": please use "On note" instead of "By the way".

(Answer): Thank you very much for your comment. According to your comment, we have changed "By the way" to "On note" in the 2.2 Patient Preparation Section as follows (Page 2):

     “On note, all current PET/MRI systems operate at a 3-Tesla static magnetic field strength.”

  1. Page 5 - "Therapy Response Assessment", 2nd paragraph: Please replace "in responders" with "in patients who have responded to therapy".

(Answer): According to your comment, we have changed "in responders" to "in patients who have responded to therapy" in the 3.3 Therapy Response Assessment Section as follows (Page 7):

     “An early study with 9 patients by Jena et al. reported that PET metabolic parameters, such as maximum standardized uptake value (SUVmax), and DCE-MRI pharmaco-kinetic parameters, such as Ktrans (volume transfer constant between blood plasma and interstitial space), obtained from simultaneous 18F-FDG PET/MRI were reduced after chemotherapy in patients who have responded to therapy

3. (To reviewer): we underwent English editing to enhance its clarity and readability.

Reviewer 2 Report

Comments and Suggestions for Authors

This is an interesting narrative review on the role of PET/MR imaging in breast cancer. Even if not particularly new, the fact that the Authors tried to include "new" tracers is a worthy effort to make this paper suitable for publications.
No particular issues are present and therefore my only comment is the fact that some clinical pictures and/or cases could be included.

Author Response

This is an interesting narrative review of the role of PET/MR imaging in breast cancer. Even if not particularly new, the fact that the Authors tried to include "new" tracers is a worthy effort to make this paper suitable for publication.

  1. No particular issues are present and therefore my only comment is the fact that some clinical pictures and/or cases could be included.

(Answer): Thank you very much for your comment. According to your comment, we have added four clinical cases as follows:

(Page 5)

Figure 2. 18F-FDG PET/MRI for initial N staging in a patient with breast cancer. (A) PET images revealed hypermetabolic right breast cancer with right axillary lymph node metastasis (red arrow). Additionally, inconclusive mild hypermetabolic uptake was found in the right interpectoral area (yellow arrow). (B) PET/MRI identified right interpectoral lymph node metastasis with gadolinium-contrast enhancement on T1-w MRI (yellow arrow).

(Page 6)

Figure 3. 18F-FDG PET/MRI for initial M staging in a patient with breast cancer. (A) PET images revealed hypermetabolic left breast cancer with left axillary lymph node metastases (red arrow). Additionally, inconclusive mild hypermetabolic uptake was observed in the sternum area (yellow arrow). (B) PET/MRI identified sternum metastasis with gadolinium-contrast enhancement on T1-w MRI (yellow arrow).

(Page 8)

Figure 4. 18F-FDG PET/MRI in a patient with previously treated breast cancer. (A) PET maximum intensity projection image revealed hepatic metastases. (B, C) Post-contrast MRI found an additional hepatic metastasis without hypermetabolism on PET (yellow arrow). (D, E) MRI ADC and PET images detected additional asymptomatic brain metastasis (red arrow). Reprinted through the Copyright Clearance Center's RightsLink® service from Kwon et al. [50].

(Page 9)

Figure 5. 18F-FDG and 64Cu-DOTA-trastuzumab PET maximum intensity projection images in a patient with recurrent left breast cancer with pulmonary metastases. (A) 18F-FDG PET detected hypermetabolic recurrent left breast cancer (yellow arrow) and pulmonary metastases (red arrow). (B) 64Cu-DOTA-trastuzumab PET demonstrated increased uptake only in the recurrent left breast lesions (yellow arrow), not in the pulmonary metastases. (C) Following treatment with trastuzumab emtansine, the metabolic activity of the recurrent left breast lesions decreased (yellow arrow), while the pulmonary metastases worsened on 18F-FDG PET (red arrow).

2. (To reviewer): we underwent English editing to enhance its clarity and readability.

Reviewer 3 Report

Comments and Suggestions for Authors

The authors performed a narrative review focused on PET/MRI and novel radiopharmaceuticals in breast cancer.

Whereas the paragraph on PET/MRI including the applications of FDG PET/MRI is well organized, I believe that the section on novel targets is not related to PET/MRI but it is related to PET/CT. The authors should clarify. I would suggest to focus on PET/MRI only.

A discussion paragraph is missing and I would suggest to add it discussing more in depth the advantages and limitations of PET/MRI ve0rsus PET/CT and other imaging methods.

Some figures on PET/MRI are needed.

Author Response

The authors performed a narrative review focused on PET/MRI and novel radiopharmaceuticals in breast cancer.

  1. Whereas the paragraph on PET/MRI including the applications of FDG PET/MRI is well organized, I believe that the section on novel targets is not related to PET/MRI but it is related to PET/CT. The authors should clarify. I would suggest to focus on PET/MRI only.

     A discussion paragraph is missing and I would suggest adding it discussing more in depth the advantages and limitations of PET/MRI versus PET/CT and other imaging methods.

(Answer): Thank you very much for your comment. In response to your comment, we have added a Discussion section, clarifying that the sections on novel targets are mostly related to PET/CT, not PET/MRI. Furthermore, we have included more comparisons and discussions of PET/MRI versus PET/CT and other imaging methods in several sections, including the Discussion section, as outlined below:

(Page 11)

"5. Discussion

     Compared with traditional imaging methods and PET/CT, PET/MRI may offer advantages for anatomical definition, particularly in certain parts of the body, including the brain, neck, breast, liver, pelvis, and bone. Diagnostic confidence can be increased with the complementary information provided by PET/MRI. Furthermore, dedicated prone breast PET/MRI, combined with supine whole-body imaging, offers more valuable information for evaluating breast cancer than supine whole-body-only imaging.

     In a prospective cohort study of 56 patients with newly diagnosed and therapy-naïve breast cancer from two university hospitals, Kirchner et al. suggested that staging by 18F-FDG PET/MRI resulted in a difference in treatment compared to the traditional staging algorithm (mammography, breast US, chest plain radiography, bone scintigraphy, US of the liver and axillary fossa, and follow-up with CT and/or MRI) in 8/56 patients (14%) [79]. For example, while the primary tumor was regarded as T2 by mammography/US, PET/MRI showed infiltration of the skin, which was confirmed by histopathology later, and the primary tumor was rated as T4. Therefore, a mastectomy and radiotherapy of the chest wall were recommended instead of breast-conserving therapy without radiation. Regarding the therapy recommendation of regional lymph nodes, differences between the traditional staging algorithm and the 18F-FDG PET/MRI were present in 6/56 patients (11%). In these patients, axillary dissection was recommended for lymph nodes in 5 cases (83%) instead of SNB based on the 18F-FDG PET/MRI that were not visible in the US. As a result, four cases were histopathologically confirmed as malignant. However, in one other case, 18F-FDG PET/MRI had a false-negative result for axillary lymph node metastasis compared with the traditional staging algorithm. A trend toward higher staging by 18F-FDG PET/MRI was found without statistical significance.

     Since PET/MRI is used less commonly in current clinical practice than PET/CT, most studies reviewed in this article regarding novel PET radiotracers for breast cancer beyond 18F-FDG were conducted with PET/CT rather than PET/MRI. Additional studies are needed to provide evidence of the benefits of using PET/MRI, instead of PET/CT, for the novel PET radiotracers in the management of breast cancer."

4.1. Estrogen Receptor (ER) section, 3rd paragraph (Page 9)

"Compared with 18F-FDG, 18F-FES does not accumulate at sites of inflammation or degeneration [55]. This property allows differentiation between ER-positive cancer/metastasis and other findings, such as inflammatory lymph nodes, degenerative osseous lesions, or reactive bone marrow. If 18F-FES PET yields a positive result for an in-determinate finding on a prior 18F-FDG PET, it can be assessed as ER-positive breast cancer/metastasis, supporting treatment decisions. In cases where 18F-FES PET shows a negative result for an indeterminate 18F-FDG PET finding, various differential diagnoses, such as benign etiologies, ER-negative breast cancer/metastasis, and other malignant etiologies, should be considered [56].

     For patient preparation, unlike 18F-FDG, 18F-FES is not significantly affected by dietary state or exercise for uptake. Thus, fasting and avoidance of strenuous exercise before imaging are not required [54]. A clinical history of any previous treatment that blocks the ER and reduces the uptake of 18F-FES, such as tamoxifen and fulvestrant, should be provided before the 18F-FES PET. The withdrawal of ER antagonists is recommended for at least 6 weeks. However, since ER-targeting drugs have various half-lives and washout times, more data may be necessary to estimate the appropriate withdrawal period [52]."

4.2. Human Epidermal Growth Factor Receptor-2 (HER2) section, 2nd paragraph (Page 10)

"Therefore, while patients with HER2-zero metastatic breast cancer are not eligible for T-DXd, HER2 PET could visualize some HER2-low lesions in these patients, making them possibly eligible for T-DXd. Additionally, HER2 PET could be a marker for predicting the efficacy of T-DXd among patients who are currently eligible for this treatment by identifying those who do not benefit from T-DXd."

4.3. Fibroblast Activation Protein (FAP) section, 3rd paragraph (Page 10)

"Backhaus et al. reported that 68Ga-FAPI-46 PET/MRI showed strong uptake in all 18 untreated primary invasive breast cancers, even for subcentimeter lesions. For axillary lymph node staging, strong uptake was found in all 13 patients with lymph node metastases. PET/MRI also detected extra-axillary LN involvement in 7 patients and affected therapy decisions in 3 patients [68]. In a later report by Backhaus et al., 68Ga-FAPI-46 PET/MRI could classify breast response to NCT correctly in all 13 patients using readers' visual assessment or TBR ratio. Evaluation of MRI alone resulted in at least 2 false positives. For lymph nodes, PET/MRI had at least 2 false-negatives, whereas MRI alone resulted in 2 false-negatives and 1 false-positive [69]."

  1. Some figures on PET/MRI are needed.

(Answer): According to your comment, we have added four Figures as follows:

(Page 5)

Figure 2. 18F-FDG PET/MRI for initial N staging in a patient with breast cancer. (A) PET images revealed hypermetabolic right breast cancer with right axillary lymph node metastasis (red arrow). Additionally, inconclusive mild hypermetabolic uptake was found in the right interpectoral area (yellow arrow). (B) PET/MRI identified right interpectoral lymph node metastasis with gadolinium-contrast enhancement on T1-w MRI (yellow arrow).

(Page 6)

Figure 3. 18F-FDG PET/MRI for initial M staging in a patient with breast cancer. (A) PET images revealed hypermetabolic left breast cancer with left axillary lymph node metastases (red arrow). Additionally, inconclusive mild hypermetabolic uptake was observed in the sternum area (yellow arrow). (B) PET/MRI identified sternum metastasis with gadolinium-contrast enhancement on T1-w MRI (yellow arrow).

(Page 8)

Figure 4. 18F-FDG PET/MRI in a patient with previously treated breast cancer. (A) PET maximum intensity projection image revealed hepatic metastases. (B, C) Post-contrast MRI found an additional hepatic metastasis without hypermetabolism on PET (yellow arrow). (D, E) MRI ADC and PET images detected additional asymptomatic brain metastasis (red arrow). Reprinted through the Copyright Clearance Center's RightsLink® service from Kwon et al. [50].

(Page 9)

Figure 5. 18F-FDG and 64Cu-DOTA-trastuzumab PET maximum intensity projection images in a patient with recurrent left breast cancer with pulmonary metastases. (A) 18F-FDG PET detected hypermetabolic recurrent left breast cancer (yellow arrow) and pulmonary metastases (red arrow). (B) 64Cu-DOTA-trastuzumab PET demonstrated increased uptake only in the recurrent left breast lesions (yellow arrow), not in the pulmonary metastases. (C) Following treatment with trastuzumab emtansine, the metabolic activity of the recurrent left breast lesions decreased (yellow arrow), while the pulmonary metastases worsened on 18F-FDG PET (red arrow).

  1. (To reviewer): we underwent English editing to enhance its clarity and readability.

Round 2

Reviewer 3 Report

Comments and Suggestions for Authors

Tha authors revised the manuscript taking into account the comments of this reviewer.